# GateLoop: Fully Data-Controlled Linear Recurrence for Sequence Modeling

## Abstract

Linear Recurrence has proven to be a powerful tool for modeling long sequences efficiently. In this work, we show that existing models fail to take full advantage of its potential. Motivated by this finding, we develop GateLoop, a foundational sequence model that generalizes linear recurrent models such as S4, S5, LRU and RetNet, by employing data-controlled state transitions. Utilizing this theoretical advance, GateLoop empirically outperforms existing models for auto-regressive language modeling. Our method comes with a low-cost $O(l)$ recurrent mode and an efficient $O(l \log_2 l)$ parallel mode, where $l$ is the sequence length, making use of highly optimized associative scan implementations. Furthermore, we derive an $O(l^2)$ surrogate attention mode, revealing remarkable implications for Transformer and recently proposed architectures. Specifically, we prove that our approach can be interpreted as providing data-controlled relative-positional information to Attention. While many existing models solely rely on data-controlled cumulative sums for context aggregation, our findings suggest that incorporating data-controlled complex cumulative products may be a crucial step towards more powerful sequence models.

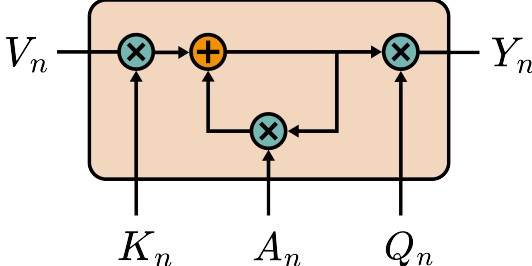

Figure 1: The GateLoop framework takes input-dependent values $V$, keys $K$, queries $Q$ and state-transitions $A$. At each step of the recurrence, the loop's input, hidden state and output is gated. While S4, S5, LRU or RetNet forget at a fixed decay rate, the fully data-controlled approach allows for input-dependent incorporation of new information, retention of memories and forgetting.

## 1 Introduction

Modeling sequences across different modalities containing long-range dependencies is a central challenge in machine learning. Historically, Recurrent Neural Networks (RNNs) have been the natural choice for this task and led to early breakthroughs in the field. However, RNNs suffer from the vanishing and exploding gradient problem, often making them unstable to train on long sequences (Hochreiter & Schmidhuber (1997)). Gated variants such as LSTM and GRU were developed to address this issue but are still inherently inefficient to train due to their non-linear recurrent nature. Furthermore, their sequential nature leads to an inductive bias towards recent inputs, limiting their practical ability to draw long-range dependencies. This inspired the attention mechanism (Garg et al. (2019)), which was first introduced as an addition to RNN for language translation, allowing the model to draw pairwise global dependencies between input data points.

Vaswani et al. (2023) took this further with Transformer, which completely gets rid of recurrence and just relies on attention. The main advantages of Transformers are their efficient parallelizable training on modern hardware and their ability to draw global pairwise dependencies. The latter property comes at the price of quadratic complexity $O(l^2)$ compared to the linear complexity $O(l)$ of RNNs. This poses a practical bottleneck for many applications, for instance limiting the document length a transformer based language model can perform reasoning on. Therefore, much effort has been put into finding attention replacements with improved complexity. While these variants such as Reformer, Linformer and Performer offer a reduced complexity of $O(l \log l)$ or $O(l)$ the original transformer with only minor adjustments prevailed due to its stronger practical performance. Furthermore, the departure from recurrence eliminated the locality bias of the model to pay more attention the recent inputs. While the absence of this bias is advantageous for some tasks, it has proven to be disadvantageous for others. This led to a line of work dedicated to injecting locality bias into Transformer (Ma et al. (2023), Huang et al. (2023)).

Meanwhile, the works of Gu et al. (2022) on the initialization of discretized State Space Models (SSMs) lead to a resurgence of linear RNNs for modeling long sequences. The most prominent model of this class S4 and its simplified diagonal variant S4D, achieve remarkable results on the long-range Arena (LRA) (Tay et al. (2020)), a benchmark designed to test a models ability to model long-range dependencies. SSMs can be trained efficiently by exploiting their linear and time-invariant nature. By rewriting the linear recurrence as a long convolution, it can be computed through the Fourier domain in $O(l \log l)$ time complexity. Smith et al. (2023b) introduced S5, which further simplifies the application of SSMs and popularized the use of associative scan implementations for fast parallelized training.

Still, SSMs are heavily dependent on involved initialization schemes. Motivated by the question whether such tedious initialization is really necessary, Orvieto et al. (2023) developed the Linear Recurrent Unit (LRU) which is on par with S4, S4D and S5 while only requiring much simpler initialization.

**Our contributions to this line of work are three-fold:**

- We show that existing models only utilize a special case of linear recurrence. Motivated by this observation, we develop GateLoop, a foundational sequence model that generalizes existing linear recurrent models by utilizing data-controlled gating of inputs, hidden states and outputs. GateLoop can be trained efficiently in $O(l \log l)$ making use of highly optimized associative scan implementations.

- Furthermore, we derive an equivalent $O(l^2)$ mode which links GateLoop to Transformer and prove that our approach can be interpreted as providing data-controlled relative-positional information to attention.

- Finally, we demonstrate the empirical effectiveness of our approach. Specifically, our results show that GateLoop outperforms the state of the art models Transformer, Hyena (Poli et al. (2023)) and S5-Hyena (Smith et al. (2023a)) on the WikiText103 benchmark for auto-regressive language modeling.

## 2 PRELIMINARIES

We consider the task of approximating sequence-to-sequence mappings. The model takes a multi-channel input sequence $x = \{x_1, \ldots, x_l\}$ packed as a matrix $X \in \mathbb{R}^{l \times d_x}$ and outputs $Y \in \mathbb{R}^{l \times d_y}$. A common assumption in this context is causality, implying that for modeling $y_n$, only information from all $x_m$ with $m \leq n$ may be used. This enables efficient training strategies such as auto-regressive language modeling.

## 2.1 RECURRENT NEURAL NETWORK

A Recurrent Neural Network (RNN) layer approximates a sequence-to-sequence mapping through the following recurrence relation involving learnable parameters $A \in \mathbb{R}^{d_h \times d_h}$, $B \in \mathbb{R}^{d_h \times d_x}$, $C \in \mathbb{R}^{d_y \times d_h}$ and an activation function $\sigma$.[1]

$$h_n = \sigma(Ah_{n-1} + Bx_n), \quad y_n = Ch_n \tag{1}$$

Common choices for $\sigma$ are tanh or sigmoid. If we chose $\sigma$ to be the identity function, the RNN layer becomes linear.

## 2.2 STATE SPACE MODEL

The continuous state space model (SSM) is characterized by the differential equation 2. Here, $\tilde{A} \in \mathbb{C}^{d_h \times d_h}$, $\tilde{B} \in \mathbb{C}^{d_h \times d_x}$, $\tilde{C} \in \mathbb{C}^{d_y \times d_h}$ are complex valued, the function $\Re(.)$ extracts the real part and $\tilde{h}(0)$ is defined to be 0.

$$\frac{d\tilde{h}(t)}{dt} = \tilde{A}\tilde{h}(t) + \tilde{B}x(t), \quad y(t) = \Re(\tilde{C}\tilde{h}(t)) \tag{2}$$

Moreover, $\tilde{A}$ can be diagonalized through its eigenvalue decomposition $\tilde{A} = V\Lambda V^{-1}$. In this representation, $\Lambda$ is a diagonal matrix of eigenvalues, and $V$ is the matrix of corresponding eigenvectors. Now, by absorbing $V$ and $V^{-1}$ into $\tilde{C}$ and $\tilde{B}$, respectively, we obtain the diagonalized SSM. For more details on this procedure, please see Smith et al. (2023b).

$$\bar{B} = V^{-1}\tilde{B}, \quad \bar{C} = \tilde{C}V, \quad \bar{h}(t) = V^{-1}\tilde{h}(t) \tag{3a}$$

$$\frac{d\bar{h}(t)}{dt} = \Lambda\bar{h}(t) + \bar{B}x(t), \quad y(t) = \Re(\bar{C}\bar{h}(t)) \tag{3b}$$

In order to utilize the SSMs practically for sequence modeling, they can be discretized, e.g., through the zero-order hold (ZOH), bilinear, or Euler method. Given a fixed discretization step-size $\Delta \in \mathbb{R}_+$, the ZOH method yields the linear recurrence relation

$$h_n = Ah_{n-1} + Bx_n, \quad y_n = \Re(Ch_n) \tag{4}$$

with the parameterization:

$$A = \exp(\Delta\Lambda), \quad B = \Lambda^{-1}(A - I)\bar{B}, \quad C = \bar{C} \tag{5}$$

Discretizing the state space model (4) gives a linear RNN layer (1) involving special reparameterizations of its weights. While this result is simply the solution of the ZOH method application, it is worth paying attention to its interpretability. Specifically, consider the influence of the discretization step size:

$$\lim_{\Delta \to 0}(A, B) = (I, 0) \tag{6}$$

In the limit $\Delta \to 0$, no new information enters the state space model and the hidden state remains constant. A small $\Delta$ leads to a sequence-to-sequence mapping with small rates of change, while a large $\Delta$ leads to large rates of change. It becomes clear, that the step-size has vital impact on the model's retain/forget properties. For S5, Smith et al. (2023b) define $\Delta$ as a learnable parameter vector, where the default values for initialization are logarithmically spaced from $0.001$ up to $0.1$. This is done in order to facilitate the learning of dependencies across different time scales.

Gu et al. (2022) observe that training SSMs with naive parameter initialization for the state transition $\bar{A}$ is not effective in practice. Grounded in theoretical memory compression results, they develop the HiPPO framework, which they utilize to find suitable initializations. Models of this class include S4, DSS, S4D and S5. Other initializations, which do not rely on HiPPO theory, nor on the correspondence to the continuous SSM representation have been proposed such as for LRU (Orvieto et al. (2023)) and RetNet (Sun et al. (2023)).

---

[1]For clarity, we omit the potential use of biases and skip connections throughout this paper. Furthermore, we consider $h_0$ to be 0.

**S4D:** The deterministic S4D-Lin initialization defines the diagonal state transition $\bar{a}$ at channel dimension $k$ to be $\bar{a}_k = -\frac{1}{2} + i\pi k$. Alternatively, the S4D-Inv initialization is $\bar{a}_k = -\frac{1}{2} + i\frac{l}{\pi}(\frac{l}{k+1}+1)$. Here, $\bar{a}$ is parameterized in continuous space. Through its ZOH discretization, $a$ is obtained.

**LRU:** The stable exponential initialization is defined as $a = \exp(-\exp(\alpha) + i\exp(\theta))$, where $\alpha$ and $\theta$ are learnable parameters.

**RetNet:** Sun et al. (2023) applies a fixed state transition formulation closely linked to the xPos positional embedding for transformers (Sun et al. (2022)). For this model, we have $a = \gamma \exp(i\theta)$ with the magnitude initialization $\gamma = 1 - 2^{-5-c}$, where $c$ is some positive constant.

## 3 DATA CONTROLLED LINEAR RECURRENCE

Incorporating data-control into deep learning models has proven to be highly successful for developing performant sequence models. Transformer, in its core, is built on the data-controlled linear operator implemented by attention (Massaroli et al. (2021)). Furthermore, Fu et al. (2023) show, that SSMs lack the data-control required for modeling language adequately. Based on this observation, they develop H3 which employs SSMs in conjunction with data-controlled element-wise gating. With this addition, they decrease the expressivity gap between Transformer and SSM-based-models for language modeling tasks. Inspired by these findings, we take the data-control paradigm further.

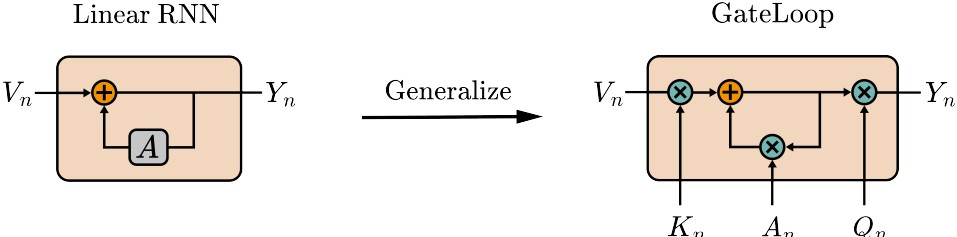

Figure 2: Omitting $B$, $C$ and application of $\Re(.)$ for clarity, we first define the input and output gates $k_n, q_n \in \mathbb{C}^{1 \times d_h}$ (row-vectors), following Sun et al. (2023). Next, as our core contribution, we replace the static state transition with content aware (diagonal) state transitions $a_n \in \mathbb{C}^{d_h \times d_h}$. This allows for time-varying control over the forget- and retention behaviour. While $q_n$ and $k_n$ act as input and output gates respectively, $a_n$ can be interpreted as a forget- and retain gate. Putting everything together, we obtain GateLoop, characterized by the the linear recurrence relation 7. We hypothesize, that allowing for time-varying control over the forget/retain behaviour can enable sequence models to keep important memories longer and discard unimportant memories faster compared to only relying on static gates. In section 5 we present experimental results which confirm this hypothesis.

$$h_n = h_{n-1}a_n + k_n^\top v_n \tag{7}$$
$$y_n = q_n h_n \tag{8}$$

For generality we define an outer product entering the gate loop leading to a hidden state $h_n$ of shape $\mathbb{C}^{d_h \times d_h}$. Choosing a (practical) max-headed variant, that is $d_h = 1$, we obtain the *SISO* case which coincides with previous definitions and element-wise gating when parallelized across multiple channels. Unfolding the recurrence relation yields equation 9, which involves a cumulative sum over preceding time steps discounted by a cumulative product of state transitions.

$$y_n = q_n \sum_{m=1}^{n} k_m^\top v_m \prod_{j=m+1}^{n} a_j \tag{9}$$

## 3.1 RELATION TO OTHER MODELS

**S4, S4D, LRU:** These models are obtained as a special case of GateLoop when not using content aware gating, nor data-controlled state transitions and only utilizing the *SISO* mode. Their defining linear recurrence relation can be unfolded into an expression which is equivalent to convolving $v$ with a structured filter. In contrast, GateLoop cannot be computed through convolution and instead we resort to associative scans for efficient computation. This is outlined in subsection 3.2.

$$y_n = \sum_{m=1}^{n} v_m a^{n-m} = (V * (\mathbf{1}_{d_h}, a, \dots, a^{l-1}))_n \tag{10}$$

**Hyena:** Poli et al. (2023) obtain a Hyena as generalization of the SSM based H3 by considering arbitrarily defined long implicit convolutions of the form $y_n = v * (K_1, \dots, K_l)$. Therefore, both GateLoop and Hyena are mutually exclusive generalizations of the linear RNN layer.

**RetNet:** Our method degenerates to RetNet when keeping data-controlled input and output gates but fixing the state transition gate.

$$y_n = q_n \sum_{m=1}^{n} k_m^\top v_m a^{n-m} \tag{11}$$

## 3.2 EFFICIENT ASSOCIATIVE SCAN COMPUTATION

Smith et al. (2023b) popularized the use of associative scan implementations for efficient parallelized computation of linear recurrence. In this subsection, we generalize their approach to derive an efficient method for computing the recurrence relation 7 for $n = 1 \dots l$ parallelized in $O(l \log_2 l)$ time complexity. Given an arbitrary associative operator $\bullet$, and a sequence of elements $\{x_n\}_{n=1}^{l}$, an associative scan computes their all-prefix sum $\Sigma$.

$$\Sigma(\{x_n\}_{n=1}^{l}) = ((x_1), (x_1 \bullet x_2), (x_1 \bullet x_2 \bullet x_3), \dots, (x_1 \bullet x_2 \bullet \dots \bullet x_l)) \tag{12}$$

The recurrence relation in 7 satisfies this form when arranging the elements $a_n$ and $k_n^\top v_n$ as the tuple leaf elements $\{x_n\}_{n=1}^{l} = \{(a_n, k_n^\top v_n)\}_{n=1}^{l}$ and defining $\bullet$ as the following.

$$p \bullet q = (p_1, p_2) \bullet (q_1, q_2) = (p_1 q_1, q_1 p_2 + q_2) \tag{13}$$

For more detailed information on prefix sum algorithms we refer to Blelloch (1990). The associative scan computes the prefix-sum efficiently in parallel through application of the binary operator on a computational tree graph. For the proof of the involved binary operator's associativity, we refer to the appendix B. In the following, we provide a simple python JAX implementation of the GateLoop operator. Note, that the parallel scan can pose a working memory bottleneck in practise for large $l \times$ nr_heads $\times d_h \times d_h$ which is why we use a max-headed variant, that is $d_h = 1$ in practise.

```python
from jax.lax import associative_scan
import jax.numpy as jnp

def gate_loop_operator(k, v, q, a):
    """
    :param k: Input gates          (l, nr_heads, d_h, 1)
    :param v: Values               (l, nr_heads, 1, d_h)
    :param q: Output gates         (l, nr_heads, 1, d_h)
    :param a: State transitions    (l, nr_heads, d_h, 1)
    """
    def binary_operator(e_i, e_j):
        a_i, kv_i = e_i
        a_j, kv_j = e_j
        return a_j * a_i, a_j * kv_i + kv_j

    kv = jnp.matmul(k, v)
    _, y = associative_scan(binary_operator, (a, kv), axis=1)
    y = jnp.matmul(q, y)
    return y
```

### 3.3 SURROGATE ATTENTION REPRESENTATION

In this subsection, we derive an mathematically equivalent surrogate attention mode for computing the recurrence in $O(l^2)$. For this, we first rewrite the cumulative product of state transitions in order to separate the variables $n$ and $m$.

$$y_n = q_n \sum_{m=1}^{n} k_m^\top v_m \left( \prod_{j=1}^{n} a_j \right) \left( \prod_{j=1}^{m} a_j^{-1} \right) \tag{14}$$

$$= \sum_{m=1}^{n} \left( q_n \prod_{j=1}^{n} a_j \right) \left( k_m \prod_{j=1}^{m} a_j^{-1} \right)^\top v_m \tag{15}$$

Using this arrangement, we can conveniently pre-compute the prefix-cumulative-product $\pi_n$ of the state transitions.

$$\pi_n = \prod_{j=1}^{n} a_j \tag{16}$$

$$y_n = \sum_{m=1}^{n} (q_n \pi_n) \left( k_m \pi_m^{-1} \right)^\top v_m \tag{17}$$

From this, the parallel $O(l^2)$ surrogate attention formulation can be obtained by packing the prefix-cumulative-product in a matrix $\Pi(A) \in \mathbb{C}^{l \times d}$ and by applying a causal mask $M \in \mathbf{R}^{l \times l}$ to the resulting surrogate attention matrix.

$$\overline{Q} = Q \odot \Pi(A) \tag{18}$$

$$\overline{K} = K \odot \Pi(A)^{-1} \tag{19}$$

$$M_{nm} = \begin{cases} 1 & n \geq m \\ 0 & n < m \end{cases} \tag{20}$$

$$Y = (\overline{Q}\,\overline{K}^\top \odot M)V \tag{21}$$

Figure 3: Considering this alternative formulation, our approach can be interpreted as providing data-controlled relative-positional information to Attention. Note, that this formulation is difficult to put into practice due to the risk of underflow during the computation of the cumulative product.

### 3.4 GENERALIZING SOFTMAX-ATTENTION

The $O(l^2)$ representation furthermore gives the opportunity of generalization for other forms of (non-linear) attention. For softmax attention this can be achieved by simply masking out the upper triangular matrix of the relative-positional-information infused attention scores with $-\infty$ and then applying softmax. The softmax sets the $-\inf$ entries to 0 resulting in the desired re-weighting of attention scores.

$$M_{-\infty}(X) = \begin{cases} X_{ij} & i \geq j, \\ -\infty & i < j \end{cases} \tag{22}$$

$$Y = \text{Softmax}(M_{-\infty}(\overline{Q}\,\overline{K}^\top))V \tag{23}$$

## 4 PRACTICAL IMPLEMENTATION

For utilizing the GateLoop framework practically, we define a simple yet powerful model. The parallel-scan computation outlined in section 3.2 was used for all experiments. To obtain values $v_n$, keys $k_n$, and queries $q_n$, we apply linear projections to the input $x_n$, following Vaswani et al. (2023). As suggested by Orvieto et al. (2023) and Sun et al. (2023), we control the magnitude and phase of the state transitions separately.

$$q_n = \text{Linear}_q(x_n), \quad k_n = \text{Linear}_k(x_n), \quad v_n = \text{Linear}_v(x_n) \quad (24)$$

$$a_n = f(\text{Linear}_\gamma(x_n)) \exp(ig(\text{Linear}_\theta(x_n))) \quad (25)$$

Inspired by the discretization of the state space model, Orvieto et al. (2023) utilizes the non-data-controlled parameterization for the magnitude $|a| = \exp(-\exp(\alpha))$, and for the phase $\arg(a) = \exp(\beta)$ where $\alpha$ and $\beta$ are model parameters. This restricts the magnitude $|a|$ to the interval $(0, 1)$ which prevents a blow-up of $a^{n-m}$ for $n \to \infty$.

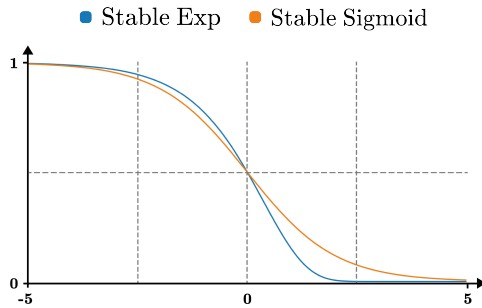

Figure 4: The stable exponential amplitude activation implemented by LRU is biased towards amplitudes close to 1. This bias is evident when plotting the (centered) stable-exponential amplitude activation function. In contrast, the sigmoid function does not have this bias. For our experiments, we chose sigmoid as the magnitude activation. Because the imaginary part of an individual state transition is not strictly required to be restricted to a specific interval, we omit the phase activation. For the model details, we refer to appendix C.

## 5 EXPERIMENTAL RESULTS

In this section, we report experimental results validating our hypothesis that data-controlled state transitions yield empirical benefits in sequence modeling. First we design a synthetic language modeling task that offers interpretable insights to our method. Moreover, we assess the performance of our method for autoregressive natural language modeling. For this we conduct experiments on the widely recognized WikiText-103 benchmark.

### 5.1 MEMORY HORIZON

Synthetic datasets are have played an important role for guiding model development, highlighting specific model advantages and weaknesses and to improve model interpretability. (Olsson et al. (2022), Fu et al. (2023)). We define our own synthetic task, specifically designed to validate the empirical advantage of data-controlled over non-data-controlled state transitions. The Memory Horizon Dataset for autoregressive synthetic language modeling is specified through an input number range, a reset token, sequence length and the number of randomized resets per sample. In order to solve this task successfully, at each time step, the past input information back to last preceding reset token needs to be memorized. We refer to appendix A for details on the underlying target compression function and dataset construction parameters. The task is designed for favoring models that can forget memories preceding an encountered reset token. Although this is a synthetic language, we hypothesize and subsequently demonstrate in section 5.2, that the fundamental capability to forget memories based on input is crucial for effectively modeling sequences from more practical modalities.

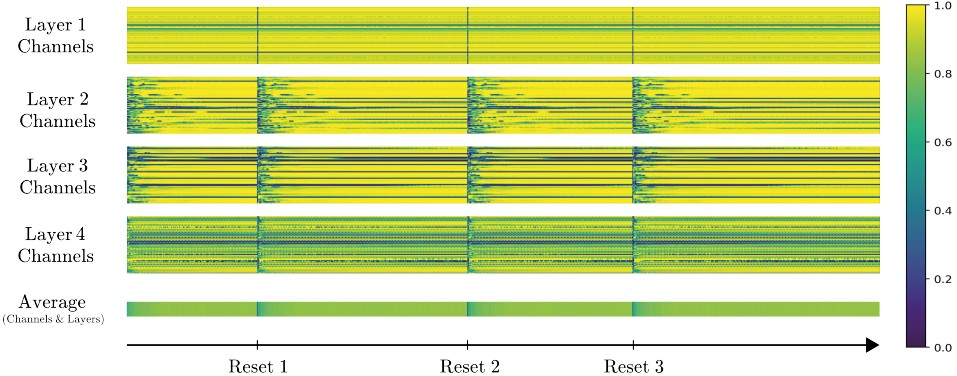

Figure 5: We visualize the applied state transition magnitudes of the trained fully data-controlled linear recurrent model, using a example sequence from the Memory Horizon dataset. Dataset details and hyperparameters can be found in appendix A and C.1 respectively. For all models layers and channels (vertically), the magnitude activations are plotted along the sequence length (horizontally). Moreover, the magnitude activation averages across channels and layers are shown. As hypothesized, through data-controlled linear recurrence, this model can learn to forget memories input-dependently by applying a (close to) zero state transition at the ideal reset positions, effectively vacating its hidden state for new relevant information.

| State transition type | Test Accuracy |
|---|---|
| Data-Controlled | **0.43** |
| Fixed | 0.25 |

Figure 6: We compare the test accuracy of the GateLoop model instance with that of a second trained linear recurrent model, which differs only in its use of a fixed state transition. The results show that making the forget/retain mechanism input dependent improves the test accuracy significantly.

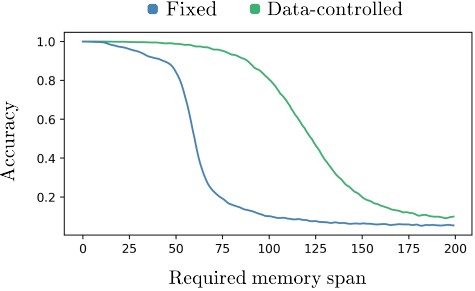

Figure 7: We plot the test accuracy over the required memory span. Not surprisingly, predicting the correct token becomes more difficult as the necessary memory capacity increases. For all required memory spans, the fully data-controlled variant performs better than the 'fixed' variant. While the performance of the latter model variant falls of rapidly after the required memory span exceeds 50, the former model variant maintains comparable performance for twice as long. Concluding, this simple synthetic language modeling task confirms that data-dependent control over the forget/retain properties can improve sequence modeling capabilities in practise.

## 5.2 WIKITEXT103

The WikiText103 dataset for autoregressive natural language modeling comprises over 100 million tokens extracted from verified Wikipedia articles. We test our fully data-controlled linear recurrent model against the state of the art competition. The model details are reported in section C.

Table 1: Comparison of WikiText103 test perplexity (lower is better) of different models. All models use the same tokenizer. The results for the other models are taken from Poli et al. (2023) and Smith et al. (2023a)

| Model | Parameters | Test Perplexity |
|---|---|---|
| Transformer | 125M | 18.6 |
| Hybrid H3 | 125M | 18.5 |
| Performer | 125M | 26.8 |
| Reformer | 125M | 26.0 |
| Linear Attention | 125M | 25.6 |
| Transformer-XL | 258M | 18.4 |
| Hyena | 125M | 18.5 |
| S5-Hyena | 125M | 18.3 |
| GateLoop | 125M | **13.4** |

GateLoop takes a significant performance leap forward over existing models while offering advantages such as avoiding softmax-attention layers (unlike Transformer and Hybrid H3), eliminating the need for tedious initialization (unlike State Space Models), and not requiring long implicit convolutions (unlike Hyena).

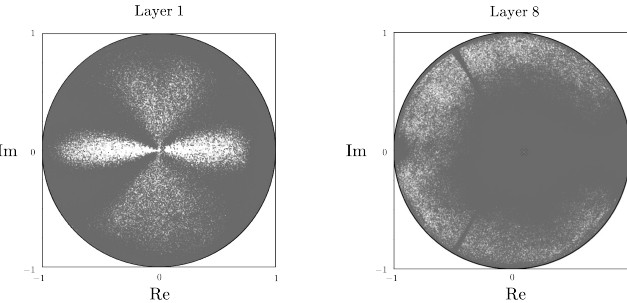

Figure 8: We plot the state transitions of the trained model for a random test input batch at layers 0 and 8. We observe structured patterns in the data-controlled state transition. While we leave interpretability for future work, we point out that these patterns indicate that the trained model deliberately utilizes the data-controlled gating of the state transition (and thus forgetting and retention of memories) by applying large varieties of magnitudes and phases.

## 6 FUTURE WORK

While our primary focus in this paper is to establish the groundwork for constructing fully data-controlled linear RNNs, we recognize the multitude of opportunities for future research. One avenue involves exploring the effects of different initialization strategies, amplitude- and phase-activations. Moreover, we suggest that future work should pay focus to the interpretability of the learned state transitions for gaining deeper insights into the model's inner workings.

## 7 CONCLUSION

We introduce GateLoop, a fully data-controlled linear RNN which generalizes existing linear recurrent models by leveraging data controlled gating of inputs and outputs and state transitions. While our method comes with linear runtime complexity $O(l)$, we derive an efficient parallelizable $O(l \log l)$ training strategy utilizing parallel scans. Furthermore, GateLoop can be reformulated in an equivalent $O(l^2)$ surrogate attention mode which reveals, that its mechanism can be interpreted as providing relative positional information to Attention. Finally we validate empirically, that fully data-controlled linear recurrence is highly performant for autoregressive language modeling.

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

## A   MEMORY HORIZON DATASET DETAILS

In this section, we describe the details of the Memory Horizon Dataset for synthetic language modeling. The goal of this dataset is to highlight the advantage of data-controlled over non-data-controlled state transitions for linear recurrent models.

Table 2: This table lists the parameters we use for constructing the Memory Horizon Dataset. The input vocabulary consists of a reset token and the number tokens for all numbers within the input number range. The output vocabulary consists of the number tokens from 0 up to the maximal output number.

| Parameter | Value |
|---|---|
| Input numbers range | $[0, 4]$ |
| Sequence length | 1024 |
| Resets per sample | 3 |
| Max output | 50 |
| Number of samples | 2000 |

Furthermore, we apply a memory compression function that computes the target token based on a list of input number tokens. This list extends from the most recent reset token to the end of the input sequence, or if no reset token is present, from the start of the sequence. The function calculates an alternating sum of products by multiplying pairs of numbers from opposite ends of the list. The operation alternates between addition and subtraction for each pair. In cases where the list has an odd number of elements, the middle element is either added or subtracted, depending on the current operation. Finally, the result is taken modulo a specified number to compress the memory value.

```python
def compress_memory(memory_list, max_output_number):
    accumulated_result = 0
    start_index = 0
    end_index = len(memory_list) - 1
    is_addition_operation = True

    while start_index < end_index:
        if is_addition_operation:
            accumulated_result += memory_list[start_index] *
            ↪  memory_list[end_index]
        else:
            accumulated_result -= memory_list[start_index] *
            ↪  memory_list[end_index]
        is_addition_operation = not is_addition_operation
        start_index += 1
        end_index -= 1

    if start_index == end_index:
        if is_addition_operation:
            accumulated_result += memory_list[start_index]
        else:
            accumulated_result -= memory_list[start_index]

    return accumulated_result % max_output_number
```

## B   PARALLEL SCAN

For completeness, we show the associativity of the utilized binary operator.

*Proof.*

$$(a \bullet b) \bullet c = (a_1 b_1, a_2 + b_2) \bullet (c_1, c_2)$$
$$= (a_1 b_1 c_1, c_1 (a_2 + b_2) + c_2)$$
$$= (a_1 b_1 c_1, c_1 a_2 + c_1 b_2 + c_2)$$
$$a \bullet (b \bullet c) = a \bullet (b_1 c_1, b_2 + c_2)$$
$$= (a_1 b_1 c_1, c_1 a_2 + c_1 b_2 + c_2)$$
$$= (a_1 b_1 c_1, c_1 a_2 + c_1 b_2 + c_2)$$

□

## C  MODEL DETAILS

**Each model layer is composed of:**

- A Time-Mixing block that aggregates information across the temporal dimension. In this case, this is the GateLoop operator with the defined content aware inputs. We use real-valued weights for the involved linear projection and return only the real part of the GateLoop output.

- A Channel-Mixing block designed to approximate functions along the channel dimension. In this experiment, a simple FNN is applied point-wise to the sequence vectors.

- Skip-Connections and Layer Normalization, which are recommended to allow information to skip channel/time mixing and stabilize training.

**The models consist of:**

- An learned input token embedding.

- A stack of $L$ model layers, with the specific number depending on the model type.

- A language head, which is a linear projection that maps the output of the last layer to a probability distribution (actually the logits) over the vocabulary. The model is trained to model the probability distribution over the possible output tokens given the current input context.

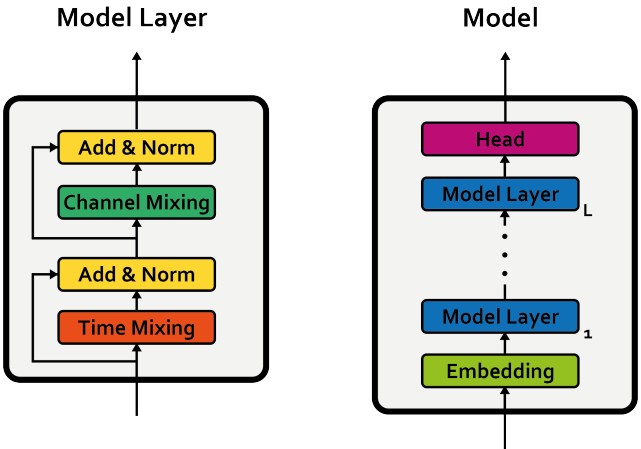

Figure 9: Visualization of the full model architecture.

## C.1 MEMORYHORIZON HYPERPARAMETERS

Table 3: Model hyperparmeters used for the MemoryHorizon experiment.

| Hyperparameter | Value |
|---|---|
| Number of epochs | 300 |
| Batch size | 32 |
| Learning rate | 0.0025 |
| Optimizer | AdamW |
| Optimizer momentum $(\beta_1, \beta_2)$ | 0.9, 0.98 |
| Weight decay | 0.05 |
| Learning rate schedule | cosine decay (linear warm-up) |
| Number of warmup steps | 10000 |
| n_layer | 4 |
| d_channel_mixing | 128 |
| d_model | 64 |
| d_qk | 64 |
| d_v | 64 |
| nr_heads | 64 |
| d_h | 1 |
| magnitude_activation | sigmoid |
| phase_activation | identity |

## C.2 WIKITEXT103 HYPERPARAMETERS

Table 4: Hyperparmeters used for the WikiText103 experiment. We apply a smaller learning to the projections which control the state transition. Moreover, no weight decay is applied to these parameters.

| Hyperparameter | Value |
|---|---|
| Number of epochs | 100 |
| Batch size | 16 |
| Base learning rate | 0.000125 |
| State transition learning rate | 0.0001 |
| Optimizer | AdamW |
| Optimizer momentum $(\beta_1, \beta_2)$ | 0.9, 0.98 |
| Weight decay | 0.25 |
| Learning rate schedule | cosine decay (linear warm-up) |
| Number of warmup steps | 5000 |
| n_layer | 12 |
| d_channel_mixing | 1872 |
| d_model | 624 |
| d_qk | 624 |
| d_v | 624 |
| nr_heads | 624 |
| d_h | 1 |
| magnitude_activation | sigmoid |
| phase_activation | identity |

