# OpenReview forum: "GateLoop: Fully Data-Controlled Linear Recurrence for Sequence Modeling"
_ICLR.cc/2024/Conference — Submitted to ICLR 2024_

### Official Review · Reviewer_kCvT · 2023-10-27

**Soundness:** 2 fair
**Presentation:** 2 fair
**Contribution:** 1 poor
**Rating:** 3
**Confidence:** 4

**Summary:**

The paper introduces a novel model (GateLoop) for long-sequence modeling. The model employs a state space model (SSM) with data-controlled state transitions to incorporate the inputs at multiple steps, hence overcoming known issues of standard linear recurrent models in language modeling (such as associative recall). To increase efficiency, they propose a parallelized version of the proposed method using associative scan computation, which improved complexity to $O(l \log l)$, where $l$ is the sequence length. The model is parametrized according to recent developments in signal propagation for linear recurrent units, i.e. by parametrizing the state transitions in polar form. Experimental evidence of the effectiveness of the method is provided on WikiText103.

**Strengths:**

1. Relevance. The paper addresses the recently growing interest in finding alternatives to Transformers for language modeling, which has been shown to underperform in tasks involving very long sequences. Hence, the proposed method can be seen as a step toward improving and generalizing this class of models.
2. Preliminary experiments are promising, outperforming existing baselines.
3. The exposition is overall clear, and the paper is well-written, with a clear story and motivation.

**Weaknesses:**

1. One of my main concerns is the experimental evidence provided and its methodology.

    a. **Evidence**: In the abstract, it is stated that "GateLoop empirically outperforms existing models for auto-regressive language modeling", but the evidence is provided on only one task. I believe that for a paper with no theory that claims practical advantages, empirical verification should be exhaustive.

    b. **Methodology**. The baselines are reported from an existing paper, and it is unclear if the comparison is fair. For instance, are the models trained with the same number of steps? Were the hyperparameters tuned consistently with the reported baselines?
2. **Motivation**. In its current form, the paper does not provide enough justification for the proposed method. Why should it perform better than H3 and its variants? The differences to H3 should be made clearer. H3 also has multiplicative interaction to effectively compare the representations of different tokens in the sequence. The authors only say that "[H3] employs SSMs in conjunction with data-controlled element-wise gating". It would be helpful if the differences were made more explicit.
Minor:
3. The model's description in section 3 should be made clearer. In particular, it is my understanding that the $k_n$ and $v_n$ values depend both on the input and weight matrices. That should be made more explicit. This can be also done for instance by outlining the algorithm/pseudo code of the forward pass of the proposed layer.
4. The quantity $l$ is used without definition in the abstract.
5. works of Gul et al --> work of. "existing only utilize a special case". "An Recurrent Neural Network". Missing reference: "we instead resort to an associative scan outlined in section?"

**Questions:**

See weaknesses.

---

### Official Review · Reviewer_BbYv · 2023-10-28

**Soundness:** 1 poor
**Presentation:** 2 fair
**Contribution:** 2 fair
**Rating:** 3
**Confidence:** 4

**Summary:**

The paper proposes a method to introduce time-variance in gated-recurrence models (GSS, H3, Hyena). The authors compute a surrogate attention mode to highlight the structure of the resulting data-controlled operator, and evaluate performance on small-scale language modeling (Wikitext103).

**Strengths:**

The idea is simple and clearly contextualized in the landscape of gated-recurrence or gated-convolution models.

**Weaknesses:**

The technical contribution is limited, and the experiments incomplete.

**Questions:**

I'll start with some minor clarifying questions:
* I find the value of 3.2 and 3.3 rather limited. For 3.2 specifically, time-variance does not affect parallel-scan compatibility with recurrences. See e.g, section 1.4.1 in Blelloch's "Prefix Sums and Their Applications", where the state-to-state parameters $a_i$ are already specified with a dependence on time. Can the authors elaborate on what is new or interesting about their exposition? (other than the diagram, which I appreciate!). For 3.3, it would have been useful to visualize the resulting surrogate attention matrices learned in some synthetic or language tasks. Could you elaborate on why you think this specific reweighting of queries and keys, resulting from the underlying time-varying system, is an appropriate parametrization?
* Is your surrogate attention matrix complex? Do you find any meaningful performance difference compared to a time-varying system with real states? How does the fact your parallel scan operates on a complex-valued sequence affect performance? It would be useful to see these comparisons.
* You use a sigmoid to parametrize the magnitude of the poles of the system. Can you motivate this choice further, ideally with an ablation study? I am not convinced the quality difference would be significant.
* The results in Table 1 are strange, especially given the fact that the authors did not rerun the baseline numbers. The perplexity gap is larger, ~13 perplexity is generally too low for small (125M) models on wikitext. As this is the only experiment, can you provide more details? Model depth, width, parametrization, learning rates, batch size, sequence length...
* How does this model compare to other gated recurrences with time-variance (e.g., GRUs, LSTMs).
* There are broken links and references e.,g section ? just before equation 9.

In summary: the idea to introduce time-variance in this way is simple and (potentially) effective. The experiments are, however, incomplete, and the novelty limited.

---

### Official Review · Reviewer_3RS5 · 2023-10-30

**Soundness:** 1 poor
**Presentation:** 2 fair
**Contribution:** 3 good
**Rating:** 3
**Confidence:** 4

**Summary:**

This paper proposes a novel architecture for deep learning on sequential data, stemming from the recent SSM trend (S4, S5, LRU, RetNet, ...). The authors propose a data-controlled variant that achieves surprisingly low perplexity in language modeling with high efficiency.

**Strengths:**

I like the illustrations and the overall presentation strategy. I enjoyed the discussion, and I think this paper has great potential but needs more work. I would say it is a week away from being an excellent submission.

**Weaknesses:**

The problem with this paper is that it is quite obviously half-cooked.
1) notation problems: equation 7 (the main equation) has non-matching matrix dimensions for multiplications and additions. This hurts clarity quite a bit. There is also a "section ?" which indicates authors did not have time for a proper final pass.
2) the paper is short: the content is, being generous, max three pages. While the idea is simple, it needs a proper comparison with existing architecture, especially on toy tasks, to enhance intuition. This brings me to my next point.
3) experiments: supporting the whole paper is a single experiment, a single perplexity number: 13.4. I do believe the architecture has great value and would love to see performance on LRA as well as on other language modeling tasks. This should not take much time. If you can do this before the rebuttal, then I would be very happy to revise my score.
4) Please compress the figures; the paper is too heavy to load!

**Questions:**

1) You leave to future work the interpretation of the (quite beautiful) eigenvalues plot you have. Do you have some hypotheses?
2) Did you evaluate the performance of simple memorization tasks (copy tasks)?

---

### Official Review · Reviewer_gLYa · 2023-10-31

**Soundness:** 3 good
**Presentation:** 2 fair
**Contribution:** 2 fair
**Rating:** 5
**Confidence:** 5

**Summary:**

The main contribution of this paper is introducing *data-dependent* state transitions into recent recurrent models (SSMs and RNNs) such as S4 variants and RetNet. This very simple modification allows the model to potential retain memory indefinitely, as opposed to prior models which all use a fixed temporal memory decay.

The paper also proposed a theoretical connection to attention with data-controlled positional embeddings, and small improvements to details of prior models (e.g. parameterization of recurrent transitions, making the activation symmetric instead of the doubly-exponential used in prior work).

Finally, the paper shows strong results on a standard autoregressive language modeling task with many established baselines.

**Strengths:**

1. The main method introduced is a very simple but motivated modification of prior work.
2. The background work is explained well as well as the connections to related models such as S4/S4D/S5, Hyena, and RetNet.
3. The surrogate attention mode is a nice connection. Although it could have explained the connection to RetNet better (and even the original Linear Attention paper by Katharopoulos) which has the same attention representation (up to the distinction of their $a$ transition being fixed). RetNet (and another related paper, TransNormer+) also frame their restricted setting as positional embeddings.

**Weaknesses:**

1. Only one empirical result is provided (WikiText-103). While the result is very strong, evaluations on more settings would strengthen the paper.
2. The paper does not comment on potential limitations of the method (see point 4 and 5 in Questions below).
3. The presentation feels rushed; there are several broken references. Some details of the model definition are not explained (see questions).
4. Overall the submission is sparse in content; it is substantially under the page limit, and large portions of the paper are dedicated to explaining prior work without providing new substance. For example, Section 3.2 on the efficient scan computation is a nice illustration but not appropriate as a contribution for an ICLR-quality paper. I would not consider it a generalization of prior work, as it is clear that the original scan operator introduced by S5 readily generalizes beyond the time-invariant setting; in fact, their experiment in Section 6.3 is applied directly to a time-varying setting.

**Questions:**

1. I could not find the exact definition for the way the data-controlled transition $a$ is defined, which seems important. While Figure 5 defines the activations, how are $\alpha$ and $\beta$ defined? Also what are the dimensionalities of $a_n$? It appears as if they are scalars in this paper, but it also claims to generalize prior SSMs which have matrix-valued $A_n$ transitions.
2. Was the surrogate attention mode implemented (like in RetNet), or just a theoretical connection?
3. The hyperparameters (Table 2) say the number of heads equals the model dimension, which seems to imply a head size of $1$. This seems at odds with the connection to multi-head attention -- why was this chosen? (I suspect it is related to the next question?)
4. How is the efficiency of the model compared to prior work? One of the motivations for the prior time-invariant SISO models (e.g. S4 and S4D) was incorporating a larger effective state size without materializing a larger latent state, which required convolutions. If it is true that a larger head dimension is not feasible, I think it is worth mentioning as a limitation.
5. While data-controlled transitions makes sense from an expressivity point of view -- it should be strictly more expressive than LTI models -- in practice, expressivity does not always predict performance as it is at odds with inductive bias. The original line of work on S4 models emphasized that they were actually designed for uniformly-sampled data such as audio instead of data such as language; this paper does not provide intuition for *why* and *when* one should expect GateLoop to perform better or worse than the prior models with fixed transitions.

---

### Meta-Review · Area_Chair_FvRa · 2023-12-12

**Metareview:**

### Summary

This paper introduces a novel sequence model called GateLoop that enhances linear recurrent structures like S4, S5, LRU, and RetNet by integrating data-controlled state transitions. By enabling indefinite memory retention and surpassing fixed temporal decay models, GateLoop demonstrates superior performance in auto-regressive language modeling tasks. Additionally, it unveils a theoretical link to attention mechanisms through data-controlled positional embeddings, hinting at more potent sequence models. The paper refines prior models by adjusting recurrent transition parameterization and symmetrically adjusting activations. Overall, it showcases compelling results against established baselines, highlighting GateLoop's efficacy in sequence modeling.

### Decision

The idea presented in this paper is interesting and the authors had the correct intuitions on developing this novel architecture to improve over other SSM models. However, as pointed out by the reviewers, the paper lacks enough experimental evidence to support its claims. I recommend the authors consider scaling their model on more challenging language modeling tasks and submit their work to a different venue. I also would recommend the authors check the writing of the paper to clarify some of the points that are missed by the reviews. As it stands right now, this paper is not yet ready for publication.

**Justification For Why Not Higher Score:**

The reviewers all agreed to reject this paper.

**Justification For Why Not Lower Score:**

N/A.

---

### Decision · Program_Chairs · 2024-01-16

Reject